# Anatomical Predictors of Clinical Improvement After Profundoplasty in Patients with an Occluded Superficial Artery: A Pilot Study

**DOI:** 10.3390/jcm14175938

**Published:** 2025-08-22

**Authors:** Denise Michelle Danielle Özdemir-van Brunschot, David Holzhey

**Affiliations:** 1Faculty of Health, Witten/Herdecke University, 58455 Witten, Germany; david.holzhey@helios-kliniken.de; 2Department of Vascular Surgery and Endovascular Therapy, Augusta Hospital and Catholic Hospital Group, 40472 Düsseldorf, Germany

**Keywords:** peripheral artery disease, deep femoral artery, profundoplasty, major amputation

## Abstract

**Background**: To date, it remains unclear which patients with occluded superficial femoral arteries (SFA) benefit from profundoplasty. Our hypothesis is that certain anatomic factors regarding the femoral bifurcation, e.g., the degree of stenosis of the common or deep femoral artery (CFA or DFA), length of stenosis of the DFA, or number of collaterals, can predict clinical success. **Methods**: An analysis pilot study was conducted to determine if the aforementioned anatomical features can predict clinical improvement and the need for major amputation following profundoplasty in patients with an occluded SFA. **Results**: Forty-nine patients were included in the analysis, most of whom were male (67.3%). The median stenosis of the CFA was 70%, and the mean stenosis of the DFA was 61%. During the follow-up period (mean 14.7 months), six major amputations were performed and 10 femoropopliteal or -crural bypasses were performed. The degree of DFA stenosis was a protective factor for major amputation (*p =* 0.04). The degree of DFA stenosis and the runoff score were associated with improvement of the Rutherford classification. However, in the multivariate analyses, none of the parameters were associated with the risk of amputation, need for peripheral bypass, or clinical improvement. **Conclusions**: Although this is a small and retrospective study, it suggests that a higher degree of DFA stenosis may be associated with a greater benefit from profundoplasty and a lower risk of major amputation. Further validation with larger patient populations and longer follow-up periods are necessary.

## 1. Introduction

Atherosclerosis is a systemic, progressive disease that can manifest as cerebrovascular disease, coronary artery disease, and peripheral artery disease (PAD). Over the past decades, new treatment options have become available. Nevertheless, there remain patients for whom revascularization is not feasible and major amputations are required. The deep femoral artery (DFA) plays a crucial role as a collateral pathway for the perfusion of the lower limbs. The DFA gives rise to the lateral and medial circumflex femoral artery, as well as multiple perforating arteries. The perforating arteries form anastomoses with branches of the popliteal artery [1,2]. In patients with an occluded superficial femoral artery (SFA), the DFA can be sufficient for limb preservation.

Due to the connections between the popliteal artery and the DFA and because DFA lesions are typically located in the proximal part, profundoplasty can be limb saving. Profundoplasty can be performed in patients in whom a femoropopliteal, -crural, or -pedal bypass is not feasible. It can also serve as an initial step in patients with combined iliac, common femoral, and superficial femoral artery disease. In such cases, iliac revascularization combined with profundoplasty can be performed first. If needed, revascularization of the femoropopliteal segment can be carried out at a later stage.

Until now, it has remained unclear which patients benefit most from profundoplasty. Identifying patients who are likely to benefit from this procedure—and distinguishing them from those for whom limb salvage is unlikely despite intervention—would greatly aid clinical decision making. The aim of this study is to analyze whether certain anatomic factors, e.g., the severity of DFA and common femoral artery (CFA) stenosis or number of communicating collaterals, can predict the effectiveness of profundoplasty in patients with an occluded superficial femoral artery (SFA). In this retrospective study, we investigated whether anatomic factors could predict clinical benefit after profundoplasty in patients with an occluded SFA.

## 2. Materials and Methods

### 2.1. Patients

We included patients who underwent profundoplasty between 1 January 2017 and 31 December 2021. Only patients with a preoperative CT angiography with a slice thickness ≤ 1 mm and a follow-up period > 30 days were included in the study. Since the measurements (e.g., artery diameter and degree of stenosis) cannot be reliably compared between CT and MR angiography, we chose to exclude patients with a preoperative MR angiography.

The included patients were divided in 2 groups:
-Group one: Patients with stenosis of the CFA and/or DFA and in whom (multiple) peripheral bypass procedures already had been performed and further peripheral bypass procedures were not possible. All these patients had an occluded SFA (TASC II D) [3]. Profundoplasty was performed as last resort for limb salvation.-Group two: Patients with an occluded SFA (TASC II D) [3] and CFA and/or DFA pathology. In these patients femoropopliteal, -crural, or -pedal bypass could be performed if profundoplasty was unsatisfactory.

In both patient groups, profundoplasty could be combined with (endovascular) treatment of inflow disease.

All data were collected retrospectively. The study was approved by the local ethics committee (University of Witten/Herdecke, S-39/2023).

### 2.2. Procedure

The indication for profundoplasty was discussed in our team, consisting of experienced vascular surgeons, and was determined on an individual basis. Generally, profundoplasty was recommended for patients with >50% stenosis of the DFA and/or CFA and SFA occlusion. For patients with Rutherford classifications 4–6, peripheral bypass surgery was preferred; when this was not possible, profundoplasty could be performed as a last resort. In cases of stenosis or occlusion of the iliac artery, a concomitant endovascular procedure of the iliac artery was recommended.

All procedures were performed under general anesthesia. A standard approach for the femoral arteries was used, with special attention to preserving the lymph nodes. The DFA was dissected until a soft segment distal to the first side branch was found [4]. Before clamping, 5.000 to 7.000 units heparin were administered intravenously. A longitudinal arteriotomy was performed from the CFA to a healthy segment of the DFA, followed by an endarterectomy. After the endarterectomy, a bovine pericardium patch was used in all patients. When profundoplasty was combined with iliac intervention, the patch was punctured, and we tried to navigate the wire in the aorta. If this approach was unsuccessful, alternative access routes such as up-and-over or transbrachial access were attempted. After crossing the lesion, the stenosis of occlusion in the iliac artery was treated with stenting.

In the postoperative phase, the patients received a single antiplatelet agent. When profundoplasty was combined with iliac intervention, temporary dual antiplatelet therapy (4 to 6 weeks) was prescribed, followed by monotherapy thereafter.

### 2.3. Predictive Factors

In all patients, the preoperative computed tomography angiography (CT angiography) was analyzed. We used a multirow 64-slice scanner, with a maximal slice thickness of 1.0 mm. The following parameters were measured: stenosis (%) of the CFA; stenosis (%) of the DFA; diameter of the DFA; number of side branches of the DFA; length of the stenosis of the DFA; number of DFA side branches exceeding 25% of the diameter of the proximal SFA; number of patent tibial arteries; and the runoff score [5].

To measure these parameters, we used dedicated software (Aycan Medical Systems, Version 3.0, Rochester, NY, USA). This software enabled semiautomatic vessel analysis and centerline composition. The maximal stenosis of the CFA and proximal DFA was calculated by dividing the luminal diameter at the site of maximal narrowing by the luminal diameter of the non-diseased arterial segment. The runoff score assesses the degree of stenosis/occlusion of the popliteal artery and the three tibial vessels [5].

Since there is no validated method to assess the quantity and/or quality of side branches of the DFA, we quantified all DFA side branches visible on CT angiography, regardless of their connection to the popliteal artery; size, or functional relevance. Additionally, the diameter or the proximal SFA was measured, and all DFA side branches exceeding 25% of the diameter of the proximal SFA, as visible on CT angiography, were assessed.

All CT angiographies were evaluated by two independent vascular surgeons, who were blinded to the clinical outcomes and the original reports. The mean of both measurements was used in further analyses. Inter-observer variability was evaluated with the intra-class correlation coefficient. An intra-class correlation coefficient of ≥0.75 was considered acceptable. When the intra-class correlation coefficient was <0.75, a third person would repeat all the measurements. Since the intra-class correlation coefficient was 0.92, no further measurements were necessary.

### 2.4. Postoperative Surveillance

Ankle brachial index was measured during hospitalization. Patients were evaluated daily. Although all decisions were made on an individual basis, common practice was to offer major amputation in group one if rest pain persisted or wounds deteriorated, whereas peripheral bypass surgery was recommended for patients in group two. Patients with Rutherford classifications 1–3 were discharged regardless of clinical improvement [6]. In an ambulatory setting, pain-free walking distance was assessed. Patients were offered surveillance at our outpatient clinic after 6–8 weeks, at 6 months, and yearly thereafter. At each visit, clinical status was evaluated and ankle brachial index was measured. We also included ankle brachial index at maximal follow-up. In some patients, bypass surgery was performed during the follow-up. For these patients, the follow-up period ended at the time of the second procedure, and clinical status and ankle brachial index before the second procedure was used. Non-compressible ankle brachial indexes were excluded from the analysis. When there was no clinical improvement or deteriorating symptoms occurred during the follow-up period, a CT angiography was performed.

### 2.5. Outcome Parameters

The reporting standards from the Society for Vascular Surgery were used to define outcomes [5]. Technical success was defined as a residual stenosis of <30% [5].

The recommended standards to define clinical status as proposed by Rutherford et al. were applied [6]:Markedly improved (ankle brachial index essentially normalized to more than 0.90);Moderately improved (no open foot lesions, still symptomatic but only with exercise and improved in at least one category, increased by more than 0.10);Minimally improved (greater than 0.10 increase, but no categorical improvement or vice versa);No change (no categorical shift and less than 0.10 change);Mildly worse (no categorical shift, but decrease of more than 0.10 or downward categorical shift with a decrease less than 0.10);Moderately worse (one category worse or unexpected minor amputation);Markedly worse (more than one category worse or unexpected major amputation).

For analysis, the first 3 categories (markedly improved, moderately improved, minimally improved) were compared with the last 4 categories (no change, mildly worse, moderately worse, markedly worse).

Major adverse limb events (MALEs) were defined as a major amputation above ankle level or major reintervention, and major adverse cardiovascular events (MACEs) were defined as myocardial infarction, stroke, or death [5]. Amputation-free survival was defined as time lived without major amputation.

Primary patency was defined as patency without the need for additional surgical or endovascular procedures, or the interval from the time of the original intervention until any intervention designed to maintain or re-establish patency was performed [5]. Assisted primary patency referred to the use of an additional surgical or endovascular procedure, as long as an occlusion of the treated site had not occurred [5]. Secondary patency was defined as patency restored after occlusion of the treated site by means of additional intervention [5].

We also analyzed the WIfI classification before and after profundoplasty. The WIfI classification covers the 3 most important parameters that put a limb a risk of amputation: wound parameter (location, size, and extensiveness), ischemia (ankle brachial index, systolic blood pressure, and toe pressure), and infection [7].

### 2.6. Statistical Analysis

Statistical analysis of the collected data was performed with SPSS 27 (SPSS Inc., Chicago, IL, USA). Continuous variables were presented as mean with standard deviations, if data were normally distributed. To asses normality, both the Kolmogorov–Smirnov test and Shapiro–Wilk test were used. When the data were not normally distributed, median and interquartile range were used. Patency rate and amputation-free survival were evaluated by Kaplan–Meier analysis. Chi-square was used for categorical data, and independent *t*-test was used for continuous data. To correct for possible confounding factors, binary logistic regression analyses were performed. Statistical significance was determined at *p* < 0.05.

## 3. Results

### 3.1. Demographics

In total, 153 patients with an occluded SFA underwent profundoplasty during the study period. We excluded patients with a pre-operative CT angiography with >1 mm slice thickness (70 patients), those with a pre-operative MR angiography (27 patients), and those with a follow-up period < 30 days (21 patients). In 50 patients, pre-operative CT angiography was performed, and the follow-up period exceeded 30 days (Appendix A).

In one patient, profundoplasty was not technically feasible. This patient had a long occlusion of the DFA and presented with critical limb ischemia; surgery was performed as a last resort. Intraoperatively, no backflow could be achieved, and the procedure was abandoned. This patient was excluded from the analysis. As a result, 49 patients were included in the study.

Most patients were male (67.3%) and presented in Rutherford classifications 3 and 4 (27.1% and 31.3%); see Table 1. The median preoperative ankle brachial index was 0.30 (interquartile range 0.00–0.50). The anatomical characteristics of the CFA and DFA are summarized in Table 2. Median stenosis of the CFA was 70% (interquartile range 40–100), and mean stenosis of the DFA was 61% (SD 49%).

### 3.2. Outcome

Median hospital stay was 7 days (interquartile range 5–18); see Table 2. Two patients died during hospital admission. One patient died due to a pulmonary embolism despite adequate prophylactic anticoagulation, and the other due to respiratory insufficiency. The latter patient had chronic obstructive pulmonary disease and was dependent on supplemental oxygen therapy. Clinical improvement was observed in 63.3% of all patients. Wound complications occurred in nine patients; in seven of these, surgical wound exploration was required, and in five negative pressure wound therapy was applied. There was no significant change in postoperative ankle brachial index. However, a significant improvement was seen in the WIfI classification, from a mean of 4.7 ± 2.0 preoperatively to 3.5 ± 2.5 postoperatively (*p* < 0.01).

Mean follow-up duration was 14.7 months (SD 23.7). During this period, sive major amputations were performed, and 10 patients underwent femoropopliteal or -crural bypass surgery. Primary patency was 88% at 1 year and remained stable thereafter. Amputation-free survival was 80% at 1 year; see Figure 1a,b. Two patients experienced a re-occlusion of the CFA and DFA; in both patients there was a simultaneous occlusion of the iliac artery.

Further analysis demonstrated that the degree of DFA stenosis was significantly associated with the risk of major amputation (*p* = 0.04); see Table 3. A higher degree of stenosis of the DFA was associated with a lower risk of major amputation following profundoplasty. The runoff score, as well as the degree of stenosis of the DFA and CFA, were identified as predictors of clinical improvement. Other anatomical features, including diameter of the DFA, length of the DFA stenosis, number of side branches of the DFA, and number of DFA side branches exceeding 25% of the SFA, were not associated with clinical outcomes.

The multivariate analyses are presented in Table 4. In the univariate analyses, the degree of DFA stenosis was a significant predictor for major amputation (*p* = 0.04) and a borderline significant predictor of improvement of the ankle brachial index (*p* = 0.05). However, in the multivariate analyses, no significant anatomical predictors of the need for major amputation, need for peripheral bypass, or clinical improvement were found.

There were significantly more patients with Rutherford classification 5 in group two (0.0% versus 23.5% *p* = 0.04), Table 1. The patients in group one had a higher degree of DFA stenosis (*p* = 0.04) and a longer length of DFA stenosis (*p* = 0.02), whereas iliac interventions were more frequently performed in group 2 (*p* = 0.02); see Table 2. Wound complications were more common in group one (*p* = 0.02). However, the risk of major amputation was comparable between the two groups (20.0% versus 8.8%, *p* = 0.27). No anatomical or clinical factors were identified that could predict the need for peripheral bypass in this patient population; see Table 3.

## 4. Discussion

The first endarterectomy was performed in the CFA by the Portuguese surgeon dos Santos in 1946 [8]. Revascularization of the DFA for the treatment of PAD was introduced in 1962 [9]. In the years that followed, outcomes varied widely: healing rates of ischemic ulcers ranged from 0 to 53%, relief of rest pain was reported in 32–94%, and 3-year limb salvage rates varied between 36 and 93% [10,11,12,13,14,15]. To date, it remains unclear which patients benefit most from profundoplasty. In current international guidelines, profundoplasty is either not mentioned at all [16,17] or only briefly addressed. One guideline described profundoplasty as “an important component of chronic limb threatening ischemia revascularization with a major impact on the long term prognosis of the limb” yet provides no specific indications or selection criteria [18]. This study was conducted to further analyze which patients benefit from DFA revascularization.

This study confirms that profundoplasty is a durable and effective procedure in patients with an occluded SFA. The technical success rate in our cohort was 98%. During the study period, only two patients experienced re-occlusion of the femoral bifurcation. Previous studies have reported primary patency rates of 80–99% [10,19], supporting the high durability of the procedure observed in our findings. A recent study also reported a mean primary patency rate of 10.7 months [20]. The limb salvage rate was 74% during a follow-up period of 3 years [20].

In this study, univariate analyses suggested that the degree of stenosis of the DFA is a protective factor for major amputation and a predictor of improvement in the ankle brachial index in patients with an occluded SFA. These findings indicate that patients with more extensive stenosis of the DFA derive greater benefit from profundoplasty, likely due to the expected improvement in postoperative blood flow. However, these results did not persist in the multivariate analyses. This discrepancy may be due to a true lack of association between these variables or could be attributed to the limited sample size, which may have reduced the statistical power of the multivariate analysis.

Although a majority of the patients (63.3%) showed an improvement in clinical status, the postoperative ankle brachial index at 30 days did not improve when compared with the preoperative ankle brachial index. A study by Wright investigated the hemodynamic effects of profundoplasty in patients with femoropopliteal occlusion [8]. The authors observed a significant increase in muscle blood flow after exercise and reported subjective clinical improvement; however, ankle brachial index remained unchanged. This was most likely due to the fact that profundoplasty does not directly improve blood flow to the tibial vessels. Over time, the ankle brachial index can improve as distal perfusion increases via collateral pathways. In our study, we observed a slight improvement of the median ankle brachial index of 0.27 at 30 days to 0.30 at the end of the follow-up period.

A recent study by Rodriguez et al. introduced a scoring system to predict outcomes after isolated CFA endarterectomy with profundoplasty [14]. The scoring system was based on three anatomic domains: degree of stenosis of the DFA, length of DFA pathology, and outflow disease. In the study by Rodriguez et al., all three of these domains were associated with clinical improvement. In this study, only the degree of DFA stenosis could predict clinical improvement after profundoplasty. Patients with a higher degree of DFA stenosis benefit the most. In six patients, major amputation was necessary despite profundoplasty; in four of them, the degree of DFA stenosis was <50%. The differences in outcome in both studies can be explained by the differences in inclusion criteria. Rodriguez et al. excluded patients with moderate to severe pathology of the SFA; in this study, only patients with TASC II D lesions of the SFA were included.

An alternative is endovascular treatment of the DFA. In particular, patients with previous surgeries can potentially benefit from endovascular surgery, as profundoplasty in these patients is associated with an increased risk of lymph-related complications, and the procedure is technically more demanding. That was also observed in our study, where patients with previous bypass grafting (group 1) had more wound complications (*p* < 0.01). A recent study described the results of endovascular therapy of the DFA [21]. This is the first study reporting a relatively large case series (*n* = 373) and a median follow-up of 40 months. DFA atherectomy was effective and showed similar mid-term clinical outcomes.

### Limitations

There are a several limitations. First, the sample size is small, with only 49 included patients. A post hoc power analysis revealed a power of 33%. More studies are therefore necessary to confirm our results. Although 153 procedures were performed during the study period, a large proportion had no preoperative CT angiography (a MR angiography was performed instead), the quality of the preoperative CT angiography was insufficient, or the follow-up period was shorter as 30 days. These patients were excluded from this study. Another limitation is the retrospective nature of the study, with its associated bias. For example, surgeons may have been more inclined to aggressively revascularize patients with a more severe lesions. Patients with more severe DFA stenosis may also have more extensive collateral pathways and therefore benefit more from revascularization.

Also, two different patient populations were studied. The first group included patients with no other surgical options, and profundoplasty was performed as a “last resort”. The second group included patients with multilevel pathology for whom profundoplasty (in 47.1% combined with iliac endovascular therapy) was the first step and peripheral bypass was still an option. However, the demographics of both groups were comparable, and separate analyses were performed. Still, there are indications that the patency of the superficial femoral artery does not influence clinical outcome [22].

## 5. Conclusions

This small, retrospective study suggests that the higher the degree of stenosis of the DFA, the more beneficial profundoplasty is. Our findings suggest that for patients with low-grade stenosis of the DFA, profundoplasty should be combined with peripheral revascularization. Further validation with larger patient populations is necessary. Also, further (randomized) studies are necessary to explore the effectiveness and safety of endovascular therapy (DFA atherectomy).

## Figures and Tables

**Figure 1 jcm-14-05938-f001:**
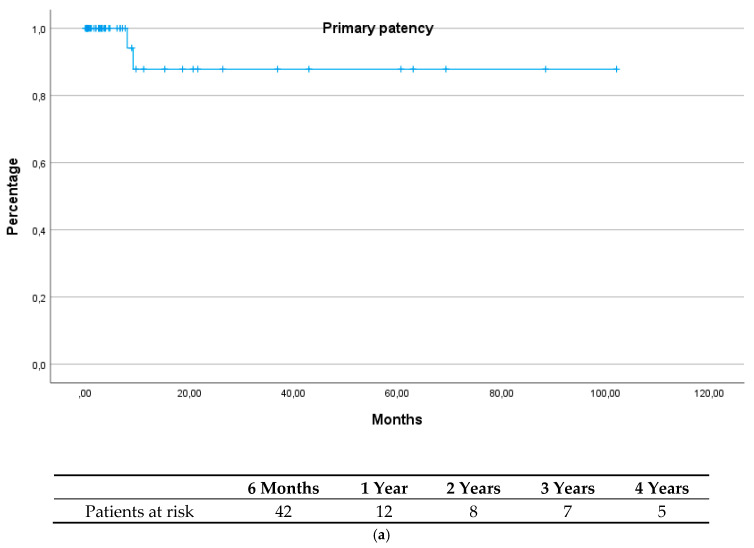
(**a**) Primary patency; (**b**) Amputation-free survival.

**Table 1 jcm-14-05938-t001:** Patient demographics.

	Total (*n* = 49)	Group 1 (*n* = 15)	Group 2 (*n* = 34)	*p*-Value
Age (years)	73.9 ± 8.5	72.9 ± 7.6	74.4 ± 9.0	0.55
Male gender	33 (67.3%)	9 (60.0%)	24 (70.6%)	0.47
Cardiovascular risk factors				
Current or former smoker	31 (63.3%)	11 (73.3%)	20 (58.8%)	0.33
Hypertension	43 (87.8%)	14 (93.3%)	29 (85.3%)	0.43
Hyperlipidemia	31 (63.3%)	7 (46.7%)	24 (70.6%)	0.11
Diabetes mellitus	16 (32.7%)	4 (26.7%)	12 (35.3%)	0.55
Diabetes mellitus, insulin dependent	7 (14.3%)	3 (20.0%)	4 (11.8%)	0.45
Comorbidities				
Coronary artery disease	18 (36.7%)	5 (33.3%)	13 (38.2%)	0.74
Cerebrovascular disease	4 (8.2%)	0 (0.0%)	4 (11.8%)	0.17
Chronic kidney insufficiency	23 (46.9%)	9 (60.0%)	14 (41.2%)	0.22
Rutherford classification *				
Rutherford 1	1 (2.1%)	0 (0.0%)	1 (2.9%)	0.25
Rutherford 2	4 (8.3%)	2 (14.3%)	2 (5.9%)	0.36
Rutherford 3	13 (27.1%)	5 (35.7%)	8 (23.5%)	0.35
Rutherford 4	15 (31.3%)	6 (42.9%)	9 (26.5%)	0.34
Rutherford 5	8 (16.7%)	0 (0.0%)	8 (23.5%)	0.04
Rutherford 6	7 (14.6%)	1 (7.1%)	6 (17.6%)	0.30
Ankle brachial index †	0.30 (IQR 0.00–0.50)	0.10 (IQR 0.00–0.36)	0.34 (IQR 0.00–0.50)	0.13
Left side	23 (46.9%)	6 (40.0%)	17 (50.0%)	0.52
Medication				
ASA	36 (73.5%)	11 (73.3%)	25 (73.5%)	0.99
Clopidogrel	6 (12.2%)	3 (20.0%)	3 (8.8%)	0.27
DOAC	12 (24.5%)	6 (40.0%)	6 (17.6%)	0.09
DOAC low dose	1 (2.0%)	0 (0.0%)	1 (2.9%)	0.50
Vitamin K antagonist	4 (8.2%)	1 (6.7%)	3 (8.8%)	0.80
Cilostazol	1 (2.0%)	1 (6.7%)	0 (0.0%)	0.13
Number of patent tibial arteries	1.8 ± 1.0	1.7 ± 1.0	1.9 ± 1.0	0.27
WIfI	4.7 ± 2.0	**	4.7 ± 2.5	

ASA = acetylsalicylic acid; DOAC = direct oral anticoagulant; IQR = interquartile range; WIfI = Wound, Ischemia, and foot Infection classification. * Rutherford classification unknown in 1 patient. † Median and interquartile range. ** Only 1 patient with Rutherford 5/6 in group 1.

**Table 2 jcm-14-05938-t002:** Anatomical features of the femoral bifurcation and outcome parameter.

	Total (*n* = 49)	Group 1 (*n* = 15)	Group 2 (*n* = 34)	*p*-Value
CT angiography measurements				
Maximal stenosis of the CFA (%) †	70 (IQR 40–100)	70 (IQR 50–100)	90 (IQR 28–100)	0.94
Maximal stenosis of the DFA (%)	61 ± 49	80 ± 41	53 ± 51	0.04
Maximal diameter of the proximal DFA (mm) †	5.1 ± 0.5	4.7 ± 0.5	5.5 ± 0.5	0.35
Length of DFA stenosis (mm)	18 (IQR 9–30)	23 (IQR 15–42)	15.5 (IQR 7.5–22.5)	0.02
Number of side branches of the DFA	5.1 ± 0.5	3.3 ± 0.5	5.7 ± 0.5	0.11
Number of side branches > 25% of the SFA	4.9 ± 0.5	3.3 ± 0.5	5.6 ± 0.5	0.08
Runoff score	12.2 ± 5.1	11.1 ± 7.2	13.0 ± 3.7	<0.01
Length of hospital stay (days) †	7 (IQR 5–18)	5 IQR 21	8 (IQR 5–17)	0.34
Concomitant endovascular treatment of iliac arteries	18 (36.7%)	2 (13.3%)	16 (47.1%)	0.02
30-day outcome				
Hematoma or major bleeding, resolved with surgical evacuation	1 (2.0%)	1 (6.7%)	0 (0.0%)	0.31
Cardiac grade 2	2 (4.1%)	0 (0.0%)	2 (5.9%)	0.34
Respiratory grade 3	1 (2.0%)	1 (6.7%)	0 (0.0%)	0.31
Wound complications, prompt recovery without surgery	2 (4.1%)	0 (0.0%)	2 (5.9%)	0.34
Wound complications, resolved with redo surgery	6 (12.2%)	5 (33.3%)	1 (2.9%)	<0.01
Pulmonary embolism, hemodynamic instability	1 (2.0%)	1 (6.7%)	0 (0.0%)	0.31
Thrombosis, resolved with redo surgery	0 (0.0%)	0 (0.0%)	0 (0.0%)	--
Ankle brachial index †	0.27 (IQR 0.00–0.53)	0.20 (IQR 0.00–0.36)	0.25 (IQR 0.00–0.56)	0.23
Mortality	2 (4.1%)	1 (6.7%)	1 (2.9%)	0.54
Midterm outcomes				
Thrombosis, resolved with redo surgery	2 (4.1%)	1 (6.7%)	1 (2.9%)	0.54
Femoropopliteal or -crural bypass	10 (20.4%)	0 (0.0%)	10 (29.4%)	0.02
Major amputation	6 (12.2%)	3 (20.0%)	3 (8.8%)	0.27
MALE	8 (16.3%)	4 (26.7%)	4 (11.8%)	0.23
MACE	5 (10.2%)	1 (6.7%)	4 (11.8%)	0.59
Rutherford classification at last follow-up *				
Rutherford 0	5 (10.9%)	1 (7.7%)	4 (12.1%)	0.51
Rutherford 1	9 (19.6%)	4 (30.8%)	5 (15.2%)	0.27
Rutherford 2	12 (26.1%)	4 (30.8%)	8 (24.2%)	0.54
Rutherford 3	5 (10.9%)	1 (7.7%)	4 (12.1%)	0.51
Rutherford 4	3 (6.5%)	0 (0.0%)	3 (9.1%)	0.33
Rutherford 5	3 (6.5%)	0 (0.0%)	3 (9.1%)	0.33
Rutherford 6	9 (19.6%)	3 (23.1%)	6 (18.2%)	0.57
Markedly improved, moderately improved, or minimally improved clinical status	31 (63.3%)	11 (73.3%)	20 (60.1%)	0.33
WIfI	3.5 ± 2.5	**	3.6 ± 2.8	
Ankle brachial index at last follow-up †	0.30 (IQR 0.00–0.62)	0.24 (IQR 0.00–0.58)	0.31 (IQR 0.00–0.65)	0.07

CFA = common femoral artery; DFA = deep femoral artery; IQR = interquartile range; SFA = superficial femoral artery; MACE = major adverse cardiac event; MALE = major adverse limb event; WIfI = Wound, Ischemia, and foot Infection classification. * Unknown in 3 patients. † Median and interquartile range. ** Only 1 patient with Rutherford 5/6 in group 1.

**Table 3 jcm-14-05938-t003:** Predictive factors of major amputation, need for peripheral bypass, and clinical improvement.

	No Major Amputation	Major Amputation	*p*-Value
Stenosis of the CFA (%)	90 (IQR 40–100)	60 (IQR 30–90)	0.33
Diameter of the DFA (mm)	5.4 ± 0.1	5.8 ± 0.8	0.18
% Stenosis of the DFA (%)	74.2 ± 35.5	45.0 ± 40.4	0.04
Length of DFA stenosis (mm)	17 (IQR 8–28)	20 (IQR 13.5–50)	0.10
Number of side branches of the DFA	6.8 ± 2.0	5.6 ± 1.6	0.47
Number of side branches > 25% of the SFA	5.3 ± 1.5	5.2 ± 1.5	0.42
Number of patent tibial arteries	1.9 ± 0.9	1.2 ± 1.2	0.05
Runoff score	12.0 ± 5.2	15.4 ± 2.2	0.07
	Need for peripheral bypass *	No peripheral bypass *	
Stenosis of the CFA (%)	90 (IQR 15–100)	90 (IQR 40–100)	0.42
Diameter of the DFA	5.3 ± 0.2	5.6 ± 0.1	0.20
Stenosis of the DFA (%)	62.0 ± 32.6	66.7 ± 41.2	0.11
Length of DFA stenosis	14.5 (IQR 7–20)	16.5 (IQR 7–30)	0.05
Number of side branches of the DFA	7.2 ± 1.8	8.2 ± 1.7	0.47
Number of side branches > 25% of the SFA	4.5 ± 1.4	5.9 ± 1.4	0.77
Number of patent tibial arteries	1.4 ± 1.1	2.0 ± 0.9	0.37
Runoff score	15.0 ± 2.9	12.2 ± 3.7	0.32
	Markedly improved, moderately improved, or minimally improved ankle brachial index	No change, mildly worse, moderately worse, or markedly worse ankle brachial index	*p*-value †
Stenosis of the CFA (%)	90 (IQR 30–100)	60 (IQR 45–100)	0.32
Diameter of the DFA	5.4 ± 0.6	5.6 ± 0.2	0.38
Stenosis of the DFA (%)	80.5 ± 37.2	49.8 ± 37.5	0.01
Length of DFA stenosis	19.6 ± 15.3	24.4 ± 19.8	0.25
Number of side branches of the DFA	7.8 ± 1.8	7.7 ± 2.2	0.95
Number of side branches > 25% of the SFA	5.5 ± 1.5	4.7 ± 1.3	0.61
Number of patent tibial arteries	1.9 ± 1.0	1.4 ± 0.9	0.81
Runoff score	11.4 ± 5.5	14.8 ± 2.8	0.02

CFA = common femoral artery; DFA = deep femoral artery; IQR = interquartile range; SFA = superficial femoral artery. * Only including patients from group 2. † Improvement versus stabile/worsening.

**Table 4 jcm-14-05938-t004:** Multivariate analyses.

	Univariate Analysis		Multivariate Analysis	
	B (95% CI)	*p*-Value	B (95% CI)	*p*-Value
Need for major amputation				
% Stenosis of the CFA	1.0 (0.97–1.02)	0.64	1.04 (0.97–1.12)	0.29
Diameter of the DFA	1.03 (0.96–1.10)	0.46	1.07 (0.91–1.23)	0.45
% Stenosis of the DFA	0.98 (0.96–0.99)	0.04	0.94 (0.87–1.00)	0.06
Length of the DFA stenosis	1.03 (0.98–1.08)	0.24	1.09 (0.99–1.12)	0.08
Number of side branches of the DFA	1.02 (0.65–1.59)	0.94	0.93 (0.46–1.87)	0.83
Number of side branches > 25% of the SFA	0.94 (0.52–1.69)	0.83	0.45 (0.11–1.86)	0.27
Number of patent tibial arteries	0.44 (0.16–1.02)	0.07	1.64 (0.13–20.61)	0.70
Runoff score	1.26 (0.97–1.71)	0.09	1.65 (0.60–4.53)	0.33
Concomitant endovascular treatment of iliac arteries	0.00 (0.00–19.76)	0.99	0.00 (0.00–25.74)	0.34
Need for peripheral bypass				
% Stenosis of the CFA	0.99 (0.98–1.01)	0.71	0.99 (0.95–1.04)	0.77
Diameter of the DFA	0.98 (0.93–1.04)	0.51	0.98 (0.89–1.08)	0.66
% Stenosis of the DFA	0.99 (0.98–1.02)	0.74	0.99 (0.95–1.05)	0.94
Length of the DFA stenosis	0.97 (0.92–1.03)	0.30	1.01 (0.91–1.13)	0.82
Number of side branches of the DFA	0.69 (0.43–1.12)	0.14	0.88 (0.45–1.72)	0.70
Number of side branches > 25% of the SFA	0.46 (0.23–0.90)	0.02	0.38 (0.15–0.99)	0.05
Number of patent tibial arteries	0.46 (0.19–1.10)	0.08	0.33 (0.07–1.61)	0.17
Runoff score	1.31 (0.99–1.73)	0.05	1.19 (0.77–1.84)	0.43
Concomitant endovascular treatment of iliac arteries	0.36 (0.08–1.75)	0.21	0.43 (0.04–5.12)	0.50
Clinical improvement as defined by Rutherford et al. [6]				
% Stenosis of the CFA	0.99 (0.98–1.01)	0.28	0.99 (0.96–1.01)	0.29
Diameter of the DFA	1.02 (0.97–1.07)	0.54	1.04 (0.96–1.11)	0.34
% Stenosis of the DFA	1.05 (1.00–1.05)	0.05	0.95 (0.93–1.01)	0.09
Length of the DFA stenosis	1.02 (0.98–1.01)	0.37	1.05 (0.99–1.11)	0.09
Number of side branches of the DFA	0.98 (0.70–1.36)	0.89	0.95 (0.62–1.46)	0.82
Number of side branches > 25% of the SFA	0.67 (0.42–1.07)	0.09	0.56 (0.30–1.05)	0.07
Number of patent tibial arteries	0.57 (0.29–1.12)	0.10	0.53 (0.20–1.43)	0.21
Runoff score	0.98 (0.85–1.00)	0.05	1.18 (0.95–1.45)	0.13
Concomitant endovascular treatment of iliac arteries	0.60 (0.16–2.23)	0.46	2.02 (0.32–12.8)	0.45

Group one versus group two.

## Data Availability

Available upon reasonable request.

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
