# Peer review of "Anatomical Predictors of Clinical Improvement After Profundoplasty in Patients with an Occluded Superficial Artery: A Pilot Study"

_jcm, 2025, doi:10.3390/jcm14175938_

Round 1
Reviewer 1 Report
Comments and Suggestions for Authors
Dear Author, I must congratulate you on the idea behind the study and the amount of data analyzed. Despite the good statistical analysis, the results were misinterpreted!
In fact, from the analysis it emerges that % of DFA stenosis is a protective factor for major amputation, not a risk factor! As you can see in Table 3, the “no amputation” group had a mean stenosis of 74.2%, versus 45% in “amputation” group. Furthermore, from the univariate analysis the B value is 0.98, which means that it is considered a “light” protective factor (because it is < 1). This result may be explained by the fact that patients with a higher degree of stenosis may benefit more from a profundoplasty. Change your manuscript and try to comment on this result in the Discussion section.
Moreover, the “form” of the manuscript should be revised:
- Line 51: To write “our hypothesis…” is not right, it is more correct to explain “aim of this study is to analyze…”
- Line 59: First, you should explain which patients you included in the study (i.e. We included patients who underwent profundoplasty between 1st January 2017 and 31th December 2021.) then which patients had been excluded, finally how you divided them into two groups.
- Explain how the indication for profundoplasty had been given (for example clinical status, % of stenosis)
- Subsequent surgeries (bypass) may be confounding and influence the results. Please consider excluding those patients from the study or limiting follow-up to the time of the second surgery
Author Response
Dear Author, I must congratulate you on the idea behind the study and the amount of data analyzed. Despite the good statistical analysis, the results were misinterpreted!
In fact, from the analysis it emerges that % of DFA stenosis is a protective factor for major amputation, not a risk factor! As you can see in Table 3, the “no amputation” group had a mean stenosis of 74.2%, versus 45% in “amputation” group. Furthermore, from the univariate analysis the B value is 0.98, which means that it is considered a “light” protective factor (because it is < 1). This result may be explained by the fact that patients with a higher degree of stenosis may benefit more from a profundoplasty. Change your manuscript and try to comment on this result in the Discussion section.
Answer: We agree with the reviewer that degree of DFA stenosis is a protective factor for major amputation. We apologize if this was not made clear in the text. We have made the necessary revisions to the manuscript (abstract page 1; results section page 11; discussion section page 14)
Moreover, the “form” of the manuscript should be revised:
- Line 51: To write “our hypothesis…” is not right, it is more correct to explain “aim of this study is to analyze…”
Answer: We thank the reviewer for this suggestion and have changed the text accordingly (introduction section page 2). Also several changed to the “form” of the manuscript were performed.
- Line 59: First, you should explain which patients you included in the study (i.e. We included patients who underwent profundoplasty between 1st January 2017 and 31th December 2021.) then which patients had been excluded, finally how you divided them into two groups.
Answer: We thank the reviewer for this suggestion and have adapted the “materials and methods” section (page 2 and 3).
- Explain how the indication for profundoplasty had been given (for example clinical status, % of stenosis)
Answer: We agree with the reviewer that this should be stated in the text and have adapted the methods section accordingly (page 3).
- Subsequent surgeries (bypass) may be confounding and influence the results. Please consider excluding those patients from the study or limiting follow-up to the time of the second surgery
Answer: We apologize that this was not stated more clearly in the methods section. We did limit the follow-up up until the time of the second surgery. The methods section was changed (page 4).
Reviewer 2 Report
Comments and Suggestions for Authors
Dear authors,
THis is a very interesting paper regarding Anatomical predictors of clinical improvement after profundoplasty in patients with an occluded superficial artery.
I have a few comments.
Results: I suggest the authors to include Wifi classification among the results section.
Discusison: I suggest the authors the following paper among discussion section:
2018 Nov;68(5):1422-1429. doi: 10.1016/j.jvs.2018.02.052. The importance of the superficial and profunda femoris arteries in limb salvage following endovascular treatment of chronic aortoiliac occlusive disease.
Abstract
Objective: This study aimed to report the long-term limb salvage, survival and patency rates of endovascular treatment for aortoiliac occlusive disease (AIOD) when outflow was achieved through the profunda femoris artery (PFA) only vs both the PFA and superficial femoral artery (SFA).
Methods: From January 2008 to July 2016, patients with AIOD who underwent aortoiliac angioplasty at the Division of Vascular and Endovascular Surgery, Hospital do Servidor Público Estadual, São Paulo, Brazil, were classified into two groups according to whether they had femoral outflow via the PFA only (group 1) or both the PFA and SFA (group 2) in the affected leg. The primary outcome was amputation-free survival. The secondary outcomes were the patency and overall survival rates.
Results: In total, 69 aortoiliac angioplasties were performed in 69 patients: 22 patients (31.8%) in group 1 and 47 (67.2%) in group 2. A total of 12 reinterventions (17.4%) were performed, seven (31.8%) in group 1 and five (10.2%) in group 2, without statistical significance between the groups (P = .063). The mean clinical follow-up period was 2500 ± 880.5 days. Both the primary and secondary patency rates analyzed at 1800 days were similar between groups 1 and 2 (80.2% vs 82.3%; P = .80 and 84.7% vs 97.6%; P = .10, respectively). Furthermore, the limb salvage rates at 1800 days were similar between groups 1 and 2 (91.3% vs 86.1%; P = .60), as were the survival rates (74.7% vs 78%; P = .80). The Bollinger score was worse in group 1 (P = .001), as expected, because of occlusion of the SFA. However, the PFA and popliteal artery scores were similar between the two groups. Occlusion of the SFA did not influence the limb salvage rate according to univariate analysis (P = .509) and multivariate Cox regression analysis (P = .671).
Conclusions: The patency of the SFA does not interfere with the outcomes of endovascular treatment for chronic AIOD. The PFA in conjunction with the popliteal artery as the sole outflow route for iliac endovascular treatment is associated with similar patency, survival, and limb salvage rates as those for outflow through both the PFA and SFA.
Author Response
THis is a very interesting paper regarding Anatomical predictors of clinical improvement after profundoplasty in patients with an occluded superficial artery.
I have a few comments.
Results: I suggest the authors to include Wifi classification among the results section.
Answer: We thank the reviewer for this suggestion and have added the Wifi classification to the methods and results section (page 5, 7-8 and table 1 and 2).
Discusison: I suggest the authors the following paper among discussion section:
Answer: We thank the reviewer for this suggestion and have changed the discussion part (page 16).
2018 Nov;68(5):1422-1429. doi: 10.1016/j.jvs.2018.02.052. The importance of the superficial and profunda femoris arteries in limb salvage following endovascular treatment of chronic aortoiliac occlusive disease.
Abstract
Objective: This study aimed to report the long-term limb salvage, survival and patency rates of endovascular treatment for aortoiliac occlusive disease (AIOD) when outflow was achieved through the profunda femoris artery (PFA) only vs both the PFA and superficial femoral artery (SFA).
Methods: From January 2008 to July 2016, patients with AIOD who underwent aortoiliac angioplasty at the Division of Vascular and Endovascular Surgery, Hospital do Servidor Público Estadual, São Paulo, Brazil, were classified into two groups according to whether they had femoral outflow via the PFA only (group 1) or both the PFA and SFA (group 2) in the affected leg. The primary outcome was amputation-free survival. The secondary outcomes were the patency and overall survival rates.
Results: In total, 69 aortoiliac angioplasties were performed in 69 patients: 22 patients (31.8%) in group 1 and 47 (67.2%) in group 2. A total of 12 reinterventions (17.4%) were performed, seven (31.8%) in group 1 and five (10.2%) in group 2, without statistical significance between the groups (P = .063). The mean clinical follow-up period was 2500 ± 880.5 days. Both the primary and secondary patency rates analyzed at 1800 days were similar between groups 1 and 2 (80.2% vs 82.3%; P = .80 and 84.7% vs 97.6%; P = .10, respectively). Furthermore, the limb salvage rates at 1800 days were similar between groups 1 and 2 (91.3% vs 86.1%; P = .60), as were the survival rates (74.7% vs 78%; P = .80). The Bollinger score was worse in group 1 (P = .001), as expected, because of occlusion of the SFA. However, the PFA and popliteal artery scores were similar between the two groups. Occlusion of the SFA did not influence the limb salvage rate according to univariate analysis (P = .509) and multivariate Cox regression analysis (P = .671).
Conclusions: The patency of the SFA does not interfere with the outcomes of endovascular treatment for chronic AIOD. The PFA in conjunction with the popliteal artery as the sole outflow route for iliac endovascular treatment is associated with similar patency, survival, and limb salvage rates as those for outflow through both the PFA and SFA.
Reviewer 3 Report
Comments and Suggestions for Authors
I had the opportunity to read the manuscript entitled “Anatomical predictors of clinical improvement after profundoplasty in patients with an occluded superficial artery”
The study aims to explore if specific anatomical features of the deep femoral artery and common femoral artery can predict the clinical outcomes after profundoplasty in patients with TASC II occlusion specifically chronic SFA occlusion. Between the 49 patients the study identifies DFA stenosis severity as a univariate predictor of major amputation risk and clinical improvement. However, these associations did not hold in multivariate analysis. The findings suggest a potential role for anatomical stratification in guiding surgical strategy, although the small cohort and retrospective design limit generalizability.
The study is well designed and addresses a specific and underexplored issue. I think it could be of great interest for the readers of the journal.
Moreover the quality of the manuscript is further enriched by the use of preoperative hight resolution CT angiography and standardized measurements that can provide precise anatomical design. Also the follow-up period with a mean of 14.7 months is adequate for early to midterm outcomes, and surveillance protocols (ABI, CT angiography when indicated) were standardized.
Here some suggestion to improve the overall quality of the manuscript:
- the final cohort of the patients consisted of only 49 patients. This could restrict the robustness of statistical testing, particularly in multivariate regression analysis, where no anatomical parameters remained significant despite their relevance in univariate analysis. Would be possible to perform a better power analysis to define required sample size for detecting clinically meaningful differences?
- The retrospective nature inherently risks bias in patient selection, particularly since patients were excluded based on imaging modality (MR angiography or duplex) or insufficient follow-up. Moreover, patients were selected for profundoplasty either as a "last resort" or as part of staged hybrid procedures, reflecting different disease severities and revascularization goals.to improve the manuscript could be useful to add a flow diagram that could detail in a more clear way the excluded patients and reasons for exclusion.
- The authors showed that patients with greater DFA stenosis had better clinical outcomes and lower amputation risk the authors treid to explore this result however the result may reflect a confounding by indication or selection artifact (e.g., surgeons may have more aggressively revascularized patients with severe DFA lesions). I think that in the discussion section this should be better explored in its potential causes (technical correction bias, dominant inflow via other pathways, etc.) or if its possible to complete the article with a propensity score matching to adjust for baseline differences between patients with varying degrees of DFA disease.
Author Response
I had the opportunity to read the manuscript entitled “Anatomical predictors of clinical improvement after profundoplasty in patients with an occluded superficial artery”
The study aims to explore if specific anatomical features of the deep femoral artery and common femoral artery can predict the clinical outcomes after profundoplasty in patients with TASC II occlusion specifically chronic SFA occlusion. Between the 49 patients the study identifies DFA stenosis severity as a univariate predictor of major amputation risk and clinical improvement. However, these associations did not hold in multivariate analysis. The findings suggest a potential role for anatomical stratification in guiding surgical strategy, although the small cohort and retrospective design limit generalizability.
The study is well designed and addresses a specific and underexplored issue. I think it could be of great interest for the readers of the journal.
Moreover the quality of the manuscript is further enriched by the use of preoperative hight resolution CT angiography and standardized measurements that can provide precise anatomical design. Also the follow-up period with a mean of 14.7 months is adequate for early to midterm outcomes, and surveillance protocols (ABI, CT angiography when indicated) were standardized.
Here some suggestion to improve the overall quality of the manuscript:
- the final cohort of the patients consisted of only 49 patients. This could restrict the robustness of statistical testing, particularly in multivariate regression analysis, where no anatomical parameters remained significant despite their relevance in univariate analysis. Would be possible to perform a better power analysis to define required sample size for detecting clinically meaningful differences?
Answer: We thank the reviewer for the thorough and thoughtful evaluation of our manuscript. We fully agree that the limited cohort size may have restricted the statistical power, particularly in the multivariate modelling, where anatomical predictors lost significance. Unfortunately, due to the retrospective nature and the highly selected population, expanding the cohort within the current study period was not feasible.
We did however, as the reviewer suggested, perform a post hoc power analysis for the univariate association between DFA stenosis and major amputation, which showed a power of 33%, which confirms a low power. This was included to the discussion part (page 15-16).
- The retrospective nature inherently risks bias in patient selection, particularly since patients were excluded based on imaging modality (MR angiography or duplex) or insufficient follow-up. Moreover, patients were selected for profundoplasty either as a "last resort" or as part of staged hybrid procedures, reflecting different disease severities and revascularization goals.to improve the manuscript could be useful to add a flow diagram that could detail in a more clear way the excluded patients and reasons for exclusion.
Answer: We agree with the reviewer and have added a supplemental figure (suppl. Figure 1). It should be noted that the numbers do not 100% correspond with the text, because in some patients there were more than 1 exclusion criteria.
- The authors showed that patients with greater DFA stenosis had better clinical outcomes and lower amputation risk the authors treid to explore this result however the result may reflect a confounding by indication or selection artifact (e.g., surgeons may have more aggressively revascularized patients with severe DFA lesions). I think that in the discussion section this should be better explored in its potential causes (technical correction bias, dominant inflow via other pathways, etc.) or if its possible to complete the article with a propensity score matching to adjust for baseline differences between patients with varying degrees of DFA disease.
Answer: We thank the reviewer for his/her valuable suggestions. We agree that our findings may be influenced by confounding factors. As the reviewer suggest, it´s plausible that surgeons more aggressively revascularized patients with more severe DFA disease, therefore improving outcomes in this group. We have changed the discussion part accordingly (page 16). A propensity score matching could not be performed due to the low number of included patients.
Reviewer 4 Report
Comments and Suggestions for Authors
Özdemir-van Brunschot and David Holzhey sought to investigate whether anatomic factors regarding the femoral bifurcation, including degree of stenosis of the common or deep femoral artery (CFA or DFA), length of the stenosis of the DFA, number of collaterals, can predict clinical outcomes in patients following profundoplasty in patients with an occluded superficial femoral artery (SFA). The authors found that that the degree of stenosis of the DFA is associated with the risk of major amputation in an univariate analysis. The study is well written, but some issues require clarification.
- This study is fully descriptive. Did the authors try to formulate a working hypothesis regarding MALE or major amputation and perform a sample size calculation based on it?
- How often did major bleeding occur after surgery? How often was a blood transfusion necessary? What was the impact of major bleeding on MALE and risk of amputation?
- How did you assess collaterals? Please provide the results of these assessments.
- Due to the small sample size the results of the study should be treated as pilot, please include this in the title and abstract.
Author Response
Özdemir-van Brunschot and David Holzhey sought to investigate whether anatomic factors regarding the femoral bifurcation, including degree of stenosis of the common or deep femoral artery (CFA or DFA), length of the stenosis of the DFA, number of collaterals, can predict clinical outcomes in patients following profundoplasty in patients with an occluded superficial femoral artery (SFA). The authors found that that the degree of stenosis of the DFA is associated with the risk of major amputation in an univariate analysis. The study is well written, but some issues require clarification.
1. This study is fully descriptive. Did the authors try to formulate a working hypothesis regarding MALE or major amputation and perform a sample size calculation based on it?
Answer: Yes, we performed a post hoc power analysis regarding need for major amputations. The power was 33%, indicating a small power and the need for more studies (page 15-16)
2. How often did major bleeding occur after surgery? How often was a blood transfusion necessary? What was the impact of major bleeding on MALE and risk of amputation?
Answer: Major bleeding was seen in 1 patient (group 1), as indicated in table 1. This patient did not develop a MALE and was not amputated.
3. How did you assess collaterals? Please provide the results of these assessments.
Answer: There is no official method to assess collaterals. We measured all visible side branches of the DFA and we measured all side branches of the DFA with a diameter > 25% of the SFA diameter (page 4). These parameters did not have a significant effect on clinical outcome (table 3 and 4).
4. Due to the small sample size the results of the study should be treated as pilot, please include this in the title and abstract.
We agree with the reviewer that this sample size is small. We have changed the title and the abstract (page 1).
Round 2
Reviewer 4 Report
Comments and Suggestions for Authors
I have no more comments.
Author Response
Thank you.